# Hepatitis C Virus Improves Human Tregs Suppressive Function and Promotes Their Recruitment to the Liver

**DOI:** 10.3390/cells8101296

**Published:** 2019-10-22

**Authors:** Laurissa Ouaguia, Olivier Moralès, Lynda Aoudjehane, Czeslaw Wychowski, Abhishek Kumar, Jean Dubuisson, Yvon Calmus, Filomena Conti, Nadira Delhem

**Affiliations:** 1Université Lille, UMR 8161–M3T–Mechanisms of Tumorigenesis and Targeted Therapies, F-59000 Lille, France; laurissa.ouaguia@efs.sante.fr (L.O.); Olivier.morales@ibl.cnrs.fr (O.M.); abhishek.kumar@ibl.cnrs.fr (A.K.); 2CNRS-UMR 8161, F-59000 Lille, France; 3Institut Pasteur de Lille, F-59000 Lille, France; 4Sorbonne Université, INSERM, Institute of Cardiometabolism and Nutrition (ICAN), F-75013 Paris, France; lynda.aoudjehane@inserm.fr (L.A.); yvon.calmus@aphp.fr (Y.C.); filomena.conti@aphp.fr (F.C.); 5Sorbonne Université, INSERM, Centre de Recherche Saint-Antoine (CRSA), F-75012 Paris, France; 6Univ. Lille, CNRS, Inserm, CHU Lille, Institut Pasteur de Lille, U1019-UMR 8204-CIIL-Center for Infection and Immunity of Lille, F-59000 Lille, France; 7Assistance Publique-Hôpitaux de Paris (AP-HP), Pitié–Salpêtrière Hospital, Department of Medical Liver Transplantation, F-75013 Paris, France

**Keywords:** HCV, HCV/JFH-1, regulatory T cells, chemokines, immune escape

## Abstract

Background: The role of regulatory T cells (Tregs) is now well established in the progression of hepatocellular carcinoma (HCC) linked to Hepatitis C virus (HCV) infection. However, nothing is known about the potential interplay between Tregs and HCV. In this pilot study, we have investigated the ability of Tregs to hang HCV on and the subsequent effect on their suppressive function and phenotype. Moreover, we have evaluated how HCV could promote the recruitment of Tregs by infected primary human hepatocytes. Methods: Tregs of healthy donors were incubated with JFH-1/HCVcc. Viral inoculation was assessed using adapted assays (RT-qPCR, Flow Citometry (FACS) and Western Blot (WB). Expression of Tregs phenotypic (CD4, CD25, CD127 and Foxp3) and functional (IL-10, GZMB, TGF-β1 and IL-2) markers was monitored by RT-qPCR, FACS and ELISA. Suppressive activity was validated by suppressive assays. Tregs recruitment by infected primary hepatic cells was evaluated using Boyden Chamber. Results: Tregs express the classical HCV receptors (CD81, CLDN1 and LDLR) and some co-receptors (CD5). HCV inoculation significantly increases the suppressive phenotype and activity of Tregs, and raises their anergy by inducing an unexpected IL-2 production. Moreover, HCV infection induces the expression of chemokines (CCL17, CXCL16, and CCL20) by primary hepatic human hepatocytes and chemokine receptors (CCR4, CXCR6 and CCR6) by Tregs. Finally, infected hepatocytes have a significantly higher potential to recruit Tregs in a seemingly CCL20-dependent manner. Conclusions: Direct interaction between HCV and Tregs represents a newly defined mechanism that could potentiate HCV immune evasion and favor intratumoral recruitment contributing to HCC progression.

## 1. Introduction

Hepatitis C is one of the major causes of chronic liver diseases worldwide [1]. The hepatitis C virus (HCV), its causative agent, is a hepatotropic, non-cytopathic, enveloped virus whose RNA genome consists of a large open reading frame encoding a ~3000 amino acid polyprotein precursor [2]. HCV entry in host cells involves main cellular receptors known as CD81, claudin-1 (CLDN1), low density lipoprotein receptor (LDLR), scavenger receptor class B member 1 (SCARB1) and occludin [3]. Some studies have also highlighted the importance of Epidermal Growth Factor (EGFR) (and CD5 co-receptors on HCV entry [4,5], suggesting that HCV may also affect immune cells.

Resolution and control of primary HCV infections are physiologically associated with a vigorous HCV-specific expansion of CD4+ and CD8+ T cells [6], those specific effector cells are able to clear the virus by non-cytolytic and cytolytic effector functions. In contrast, the development and persistence of a chronic infection are linked to a weak or an absent HCV-specific Th1 response along with the presence of Th2 cytokines (IL-4 and IL-10) [7,8]. Many studies have described a failure of both innate and adaptive immune responses partly due to CD4 and CD8 T cells failure, impaired cytokine production, altered dendritic cell function and induction of regulatory T cells [8].

Regulatory T-cells are specialized subsets of the adaptive immune system that are able to modulate the immune response of the host [9,10,11] by suppressing the auto-reactive T cells, reducing inflammation and inducing tolerance. Based on their origin and their major mechanisms of action, we can distinguish thymic-derived Tregs called “natural Treg (nTregs or Tregs)” cells and periphery-induced Tregs also known as induced Tregs (iTreg) or Type 1 regulatory T cells (Tr1). Indeed, under certain conditions, some FOXP3+ Treg cells differentiate from conventional CD4+ T cells (Tconv) during the peripheral immune response and one of their main mechanisms of action is due to the release of a large amount of the immunosuppressive cytokine IL-10 [12,13]. nTregs are the best known and they are CD4 positive T cells characterized by the constitutive overexpression of CD25 (alpha chain of the interleukin (IL)-2 receptor) surface markers and the absence or a sub-expression of CD127 (alpha chain of the IL-7 receptor) [11,14]. Their mechanisms of immune suppression depend on both cell-to-cell contact and immunosuppressive cytokine secretion [15].

It has been shown that extrinsic regulatory pathways are also involved in HCV-specific response. Indeed, the weakening of the patient’s immune response is partly due to the elevated frequency of intra-hepatic and circulating Treg [16,17]. Tregs have been suggested to contribute to HCV-specific CD8+ T cell dysfunction through IL-10 and TGF-β secretion and through expression of CD25, CLTA4 and LAG3 [18,19]. Others and we have shown that Treg and Tr1 cells are more important in patients with severe hepatitis C recurrence compared to those without or with minor recurrence [13,18,20,21]. Moreover, few studies have highlighted the role of CCL17 and CCL22 chemokines in the Treg recruitment to the liver [22]. Hence, the differentiation and/or the recruitment of regulatory T-cells to the liver are probably key factors of the disease progression. However, the mechanisms by which HCV might induce the recruitment and proliferation of Tregs remain unknown.

In this pilot study, we reported that HCV inoculation increased the proliferation, the expression of regulatory markers and the suppressive function of Tregs that are correlated with transcriptional activation and production of suppressive factors. Very interestingly, we also showed that HCV inoculation increased Treg recruitment by human primary hepatocytes in a CCL20-dependent manner, suggesting a key role of Treg in modulating the hepatic immune response to favor HCV infection.

## 2. Materials and Methods

### 2.1. Human Donors

Human blood samples were collected from healthy adult donors after obtaining informed consent in accordance with the approval of the Institutional Review Board at the Biology Institute of Lille (DC-2013-1919). Human primary hepatic cells were isolated from patients undergoing partial hepatectomy for treatment of colorectal cancer metastases. All were seronegative for HCV, HBV and HIV. The procedures for cell isolation were carried out in accordance with the approval of the CPP (Comité de Protection des Personnes) as previously described [23,24].

### 2.2. Isolation of Immune Cells and Flow Cytometry

Peripheral blood mononuclear cells (PBMC) were first isolated on Ficoll-Paque^TM^PLUS gradient (GE HEALTHCARE, Little Chalfont, UK), and both CD4+ natural regulatory T-cells (Treg) and CD4+ conventional T-cells (Tconv) were isolated from filtered PBMC, using CD4+CD25+ Regulatory T cell Isolation Kit, human (Miltenyi Biotec, Bergisch Gladbach, Germany) as previously described [25]. Tregs were counted and directly stained for CD4, CD25 and CD127 expressions with mouse anti-human CD4-phycoerythrin(PE)-cyanin(Cy)5 (BD Pharmingen, Le Pont de Claix, France), Mouse anti-human CD25-PE (Miltenyi Biotec, Bergisch Gladbach, Germany) and Mouse anti-human CD127-fluorescein isothiocyanate (FITC) (Clinisciences, Nanterre, France), according to the manufacturer’s instructions. In some experiments, Tregs, PBMC, Huh7 and or HepG2 (hepatoma cell lines) were directly stained for membranous protein receptors CD81, CLDN1, SCARB1 and LDLR. Samples were acquired on FACS Calibur flow cytometer powered by CellQuest Pro software v5.1 (Becton Dickinson, Le Pont de Claix, France) and analyzed using Flow Jo software 10.0.7 (Tree Star Inc., Ashland, OR, USA).

### 2.3. Isolation of Hepatic Cells

The liver tissue fragments were perfused, first with liver perfusion medium (Invitrogen, Carlsbad, CA, USA) at 37 °C for 15 min, and then with collagenase and dispase-containing liver digest medium (Invitrogen) at 37 °C for approximately 15 min, until the tissue was completely digested. Cells were dispersed by gentle shaking, filtered and centrifuged at a low speed. Primary human hepatocytes (PHH) were collected from the pellet and used in primary culture, as previously described [23]. The non-parenchymal cell fraction was collected from the supernatant that was centrifuged to obtain intra-hepatic fibroblasts (IHF), as described elsewhere [24,26].

### 2.4. Culture Conditions and Viral Inoculation

Infectious HCV virions produced in cell culture (HCVcc) were obtained after transient transfection of full genome (JFH-1) in Huh7 cell lines. Huh7 cells were cultured in DMEM Glutamax medium supplemented with 10% heat inactivated FBS, 1% of MEM NEAA, 100 U/mL of penicillin and 100 µg/mL of streptomycin (Complete Huh7 medium) for different days at 37 °C in different experiments. HCVcc was collected over a 1-week period, filtered (0.22-μm-pore-size filter), divided into single-use aliquots, and stored at −80 °C until further use. Isolated Treg and PBMCs cells were either stained and characterized by flow cytometry or cultured in flat bottom 48 well plates at a rate of 1.2 to 1.5 × 10^6^ cells per well and activated as described in the Appendix A and methods. Cells were incubated for 3, 24, 48 or 72 h in the presence of HCVcc/JFH-1 (multiplicity of infection (MOI) 0.4)) in complete Huh7 medium. Isolated primary hepatic cells were, in turn, infected in the same manner with MOI 0.1 for hepatocytes and MOI 1 for fibroblasts. Negative control of inoculation was carried out by cultivating Huh7 cells in complete medium. After several passages, supernatants of non-infected Huh7 cells were harvested and used as negative control in all experiments.

### 2.5. Migration Assay

Treg were harvested and resuspended in DMEM at 10^6^ cells/mL with or without HCVcc inoculation. Chemotaxis experiments were performed with the supernatant of primary human hepatic cells culture (PHH and IHF) either infected or not and in the presence or absence of blocking mAbs anti-CCL20, anti-CXCL16 and anti-CCL17 (PeproTech, Rocky Hill, NJ, USA) at different concentrations. DMEM and SDF1α (10^−7^M; PeproTech, USA) were used as negative and positive controls respectively. h-rec-CCL20 (50 ng/mL), h-rec-CCL17 (10 ng/mL) and h-rec-CXCL16 (100 ng/mL; PeproTech, USA) were also used as specific controls. Each condition was performed in triplicate. Cells that migrated through the 5 µm filter of the inferior wells were counted by at least two different investigators. Results were presented as index of migration compared with the negative control DMEM.

### 2.6. Gene Expression

Primers and reagents to quantify markers related to Treg phenotype, activity, proliferation and recruitment were obtained from different manufacturers (QIAGEN, Venlo, Nederland; Sigma Aldrich, Saint-Louis, MO, USA; Invitrogen Carlsbad, CA, USA and GE Healthcare, Little Chalfont, UK). All primers sequences are listed in the Appendix A.

### 2.7. Statistical Analysis

Results are given as mean ± SEM. Data was analyzed using Prism 6.0 software (Graph Pad Software Inc., San Diego, CA, USA). All quoted *p*-values are two-sided, with *p* ≤ *0.05* (*), *p* ≤ 0.001 (**), *p* ≤ 0.0001 (***) and *p* ≤ 0.00001 (****) being considered statistically significant for the first and highly significant for the others.

### 2.8. Supplementary Material and Methods

For more information on flow cytometry, in vitro cell activation and proliferation, Treg suppression assays, cell viability, cell lysis, protein extraction and quantification, RNA extraction and RT-qPCR analysis, please refer to Appendix A and methods.

## 3. Results

### 3.1. Characterization of Freshly Isolated Natural Human Regulatory T Cells

We first characterized the phenotype and function of freshly isolated Treg. Cytometric analysis showed that 95.8% of freshly isolated Treg are CD4^+^CD25^high^ and 94.5% are CD4^+^CD25^high^CD127*^-/low^* (Appendix A). Further analysis reveals that freshly isolated Tregs were totally anergic in vitro and significantly inhibit the proliferation of PBMC (Appendix A) principally through cell-lysis mechanisms in a dose dependent-manner (Appendix A). These results clearly showed that isolated Tregs were functional.

### 3.2. Circulating CD4^+^CD25^+/high^CD127^−/low^ Tregs Possess the Classical HCV Entry Receptors

Our first question was to highlight the presence of HCV entry receptors on isolated Tregs, and we showed that they display mRNA expression of *CD81*, claudin-1 (*CLDN1*) and low-density lipoprotein receptor (*LDLR*; Figure 1, left panels). *CD81* and *LDLR* mRNA expressions seem to be higher in Tregs compared to control Huh7 whereas scavenger receptor class B member 1 (*SCARB1)* and *CLDN1* mRNA expression are respectively lower and similarly expressed (Figure 1B,C, left panels). These results were confirmed by FACS and WB analysis that clearly indicate the presence of CD81, CLDN1 and LDLR HCV entry receptors on Tregs (Appendix A). PBMC and Huh7 cells were used as control (Figure 1, middle and right panels). Comparing to *Huh7*, the best correlation in term of protein levels can be observed for CLDN1 with an overexpression in total as well as in membranous protein forms whereas the gene expression is the same as for Huh7. For CD81, the gene expression was twice the one in Huh7 with the same observation for relative expression of total protein in WB, but the final membranous expression remains quite the same as for Huh7. For SCARB1, the gene expression was 12.5 times less in Treg then in Huh7, which was the same in term of relative expression between total proteins (WB) but a slight overexpression was finally reached at the membranous expression. Finally, for LDLR, although there was a real gene overexpression in Tregs compared to Huh7, there was a lower relative expression in term of total protein and equivalence at the membranous level. Then comparing to PBMCs, the results for CD81, SCARB1 and LDLR present a quiet good correlation between total relative expression (WB) and membranous expression levels. Indeed, in one way CD81 was over expressed when studying relative expression of total protein by WB and it was the same observation, with a slight decrease, when looking at the membranous expression by FACS. The other way was for the subexpression of SCARB1 and LDLR with almost the same relative levels according to the technique used. Finally, one could observe a difference for CLDN1 where although a subexpression was detected in term of total relative protein a slight overexpression was observed at the membranous level (Appendix A).

In contrary, HepG2 cells did not express CD81 or LDLR receptors (Appendix A). These experiments suggest that Tregs could potentially interact with HCV.

### 3.3. HCV Inoculation Increases the Expression of Its Receptors on Tregs

To visualize the impact of HCV on the expression of its receptors on CD4^+^CD25^+/high^CD127^−/low^ Tregs, we performed RT-qPCR assay comparing inoculated Tregs (cultured with HCVcc) versus non-inoculated Tregs (cultured with the supernatant of non-infected Huh7). As shown in Figure 2, the inoculation with HCVcc increased the mRNA expression of CLDN1 at 3 h post-inoculation (3 h p.i; Figure 2A). In addition, HCVcc inoculation significantly increased the gene expression of CD81, EGFR and CD5 at 24 h p.i (Figure 2B). These data suggest that HCV seems to make Tregs more sensitive to a virus interaction.

### 3.4. Supernantant Containing HCVcc Significantly Modifies the Suppressive Phenotype of Tregs

As shown in Figure 3, HCVcc inoculation significantly increased the expression of natural Tregs suppressive phenotype (Figure 3). Indeed, results showed that inoculation with HCVcc increased the mRNA expression of CD4 and FOXP3 at 3 h post-inoculation (3 h p.i; Figure 3A) and CD4, CD25, FOXP3, CTLA4 and LAG3 at 24 h p.i (Figure 3B) with at least five fold change compare to control cells. In addition, protein analyses by flow cytometry revealed a statistical increased of the percentage of positive stained cells of the suppressive phenotype CD4+CD25highCD127- within natural Tregs at 3 h p.i (34.2% vs. 13.4%) and 24 h p.i (30.4% vs. 17.2%; Figure 3C). Further analyses reveal that CD4+CD25high T cells subset were 98.525% vs. 99% and 91.275% vs. 93.050% respectively at 3 h p.i and 24 h p.i within inoculated Treg compared to control conditions (Appendix A). In addition, the percentage of positive stained cells CD25highCD127- subsets within the CD4+ population were significantly higher in inoculated Treg at 3 h p.i and 24 h p.i. (36.40% vs. 18.78% and 28.175% vs. 12.048% respectively) compared to control cells (Appendix A). There was no significant difference on the percentage of positive stained cells CD4+CD25high subsets within Treg inoculated with HCVcc compare to freshly isolated Tregs neither at 3 h nor at 24 h p.i (Appendix A). Nevertheless, there was a significant increase of CD25highCD127- subsets within the CD4+ population, 36.40% vs. 14.695% at 3 h p.i and a tendency to increase at 24 h p.i. (28.175% vs. 14.695%) in between Treg inoculated with HCVcc and freshly isolated Tregs (Appendix A). Therefore, HCV seems to be associated with a reinforcement of natural Treg suppressive phenotype.

### 3.5. HCV Inoculation Induces Treg Proliferation

We analyzed the impact of HCVcc inoculation on Tregs proliferation. Both viability assay by cell counting and (^3^H)-thymidine incorporation assays indicated that HCV inoculation effectively increases CD4+CD25highCD127−/low Treg proliferation within time (Figure 4A–C). To highlight the underlying mechanism, we performed transcriptomic analyses showing a decrease in the gene expression of IL-2 and IL-4, two molecules related to lymphocytes proliferation at 3 h p.i (Figure 4D). On another hand, there is an increase in the expression of all markers at 24 h p.i for inoculated Tregs, but only the expression of IL-2 and IL-15 are statistically significant (Figure 4E). Further experiments show an effective and sustain increase of IL-2 secretion at 48 h p.i by Tregs (Figure 4F), suggesting that HCV inoculation could favor Tregs proliferation.

### 3.6. HCV Inoculation Increases the Suppressive Activity of Tregs

We investigated here the impact of HCV inoculation on Treg suppressive function. Tregs were previously cultured with HCV particles for 24 h or cultured with supernatant of non-infected Huh7 for negative controls. Then, we made co-culture of PBMC and washed Tregs at 2:1 ratio for 48 h in activated conditions. By (^3^H)-thymidine incorporation assays, we demonstrated that HCV inoculation significantly increased the capacity of Tregs to inhibit the PBMC proliferation (Figure 5A). Mechanistic analyses reveal that HCV inoculation increases the expression of suppressive factors *IL-2RA* (CD25), *IL-35* (EBI3 sub unit), Granzyme B (*GZMB*) and *TGF-β1* at 3 h and 24 h p.i (Figure 5B,C). Furthermore, we evaluated the secretion of immunosuppressive cytokines. Results demonstrate that HCV inoculation increased the secretion of TGF-β1 at 48 h p.i by Tregs (Figure 5D) but did not change the IL-10 secretion at any time (Figure 5E). Our study also shows that HCV inoculation increased the gene expression and the effective secretion of pro-inflammatory cytokines such as IL-23 (at 3 h) and IL-17 (at 48 h) by Tregs (Appendix A). These data clearly show that HCV not only promoted the Tregs phenotype, but also increased its suppressive and harmful activity.

### 3.7. CD4^+^CD25^high^CD127^−^ Tregs are Recruited by Infected Human Hepatic Cells Supernatants

To examine whether supernatants of in vitro infected PHHs could attract more Tregs than supernatants of non-infected PHHs, Boyden chamber migration assays were performed. Natural Tregs obtained from PBMC of healthy donors were placed in the upper chamber of a Boyden system (containing a 5 µm pore size filter), while supernatants of PHH cultures were applied in the lower chamber. After 3 h, the number of Tregs attracted was estimated. As shown in Figure 6A, both supernatants induced migration of CD4^+^CD25^high^CD127^−^ Tregs but the proportion and quantity of CD4^+^CD25^high^CD127^−^ Tregs were only significantly higher when using supernatant of infected PHHs. Importantly, Treg recruitment by infected PHHs looked comparable to the one induced by the positive control SDF1ɑ (Figure 6A, grey bars) comparing the mean of migration index. Otherwise, no significant differences have been observed between non-infected PHH supernatants and control medium DMEM on Tregs recruitment (Figure 6A). In addition, when Tregs were previously cultured with HCV, there was an increase of the recruitment regardless of the supernatant (Figure 6A). Transcriptomic experiments showed that HCV inoculation of PHHs increases the expression of several chemokines such as *CCL2, CCL20* and *CXCL16* (Figure 6B). In addition, our analyses also reveal that HCV inoculation increased significantly the expression of corresponding chemokine-receptors by Tregs such as *CCR4* at 3 h p.i, CXCR4 at 24 h p.i *CCR2* and *CCR6* both at 3 h and 24 h pi, and not significantly for *CXCR6* at both time points (Figure 6C). Furthermore, we evaluated the effective secretion of three chemokines by PHHs and results showed that HCV inoculation seemed to increase the secretion CCL20 (Figure 6D, left panel) but did not really change the secretion of CCL17 and CXCL16 (Figure 6E,F, left panels). Importantly, in the chemotaxis assay, the addition of neutralizing antibodies against CCL20, CCL17 and CXCL16 to the infected PHH supernatant significantly inhibited the recruitment of Tregs (Figure 6D–F, right panels) mostly in an all or nothing principle rather than in a dose dependent manner. These data strongly suggest that HCV-infected hepatocytes promote the recruitment of Tregs to the liver. Other analyses revealed that IHFs incubated with HCVcc did not exhibit increased chemo-attractant abilities, as demonstrated by similar migration index as DMEM (Figure 7A) and no increased secretion of CCL20, CCL17 and CXCL16 upon HCV inoculation (Figure 7B). Moreover, and finally on this point one could mention that every antibody used was efficient and specific for its blocking capacity. Obviously in sight of this analysis one could mention that anti-CCL20 were very efficient regarding the high amount of the chemokine present in the PHH conditioned media and their dose response scheme either for NI or I Tregs. The other ones were also efficient with a capacity of blocking 40% of the chemo-attractiveness of the PHH conditioned media.

## 4. Discussion

Recently, attention has focused on the contribution of Tregs to hepatitis C progression [16,17,21]. Nevertheless, no study has addressed the direct impact of HCV on these cells, and the mechanisms by which HCV might increase Treg frequency or improve their recruitment to the infected liver remains unknown. In the present pilot study, we reported that in vitro HCV inoculation induced a significant increase of the Treg number and promoted their suppressive phenotype and activity. Very interestingly, we also reported that HCV infection significantly increased the recruitment of Tregs by PHHs involving CCL-20 pathway.

We have first shown that CD4^+^CD25^high^CD127*^−/low^* Tregs expressed CD81, CLDN1, LDLR and SCARB1, which have been clearly associated with HCV entry in host cells [3,27]. The levels of expression were assessed by different approaches. To make clear about the importance of the expression of these key features of the studied cell type we compared it to the Huh7 permissive cell, the gold standard for their expression. The consistency of these data were obvious for PBMCs, it was a bit less for Huh7 or Tregs, with SCARB1 presenting most of the discrepancies observed but only in term of membranous expression for Tregs and Huh7. Many studies reported discordant correlation between some subsets of protein and mRNA expression [28,29,30]. Indeed, mRNA levels are often not reflected in protein levels due to multiple layers of gene regulation like genomic variations, gene expression, protein translation and post-translational modifications [31]. For the difference with SCARB1, it can be explained by the fact that within PBMC relies one major source of SCARB1 in the monocytes/macrophages cell, which plays a major role in the lipoprotein handling [32].

Our pilot study also revealed that Tregs expressed some co-receptors like epidermal growth factor (EGFR) and CD5 (Figure 2). Interestingly, the CD5 receptor has been more recently described as the major HCV entry receptor on T lymphocytes [5,33,34]. Thus, its presence on freshly isolated human Tregs suggests that they could directly interact with HCV. Furthermore, we showed an increased gene expression of HCV entry receptors and co-receptors on Tregs, from the early stage after HCV inoculation, suggesting that HCV makes Tregs more sensitive to the virus interaction. These data correlated with studies showing that HCV successfully binds PBMCs of which the majority are T lymphocytes [35].

To investigate the potential of HCV to modulate the immune phenotype of human peripheral Tregs, we challenged them with HCV particles or with supernatant of non-infected HuH7, and we studied their phenotypic and functional changes in the activated condition. Interestingly, *CD4, CD25* and *FOXP3* expressions on Tregs were increased after HCV inoculation, which is in accordance with the finding that CD4+CD25^high^FOXP3+ Tregs constitute the most suppressive phenotype of Tregs [36]. We also found that Tregs overexpressed the inhibitory markers *CTLA-4* and *LAG3* after HCV inoculation. This is consistent with previous studies reporting that exhausted T cells expressed inhibitory markers in chronic HCV infection and that their trafficking from intracellular vesicles to the cell surface determines T cell activation [37]. Interestingly, we also recorded a concomitant decrease of CD127 expression levels on HCV-inoculated Tregs that is associated to the most suppressive Treg phenotype [38,39]. Taken together, these observations support the idea that HCV inoculation significantly increases the suppressive phenotype of Tregs.

More surprisingly, our data revealed an increase proliferation of activated Tregs. This was unexpected because nTregs are described as totally anergic in vitro, partially due to their inability to produce IL-2 proliferative factors [40]. Using transcriptomic analyses, we showed that HCV inoculation decreased the expression of *BLIMP1* and gradually increased the gene expressions of *IL-2, IL-4, BCL6* and *IL-15* closely related to T cell proliferation [41,42]. These data were confirmed by an effective secretion of the proliferative cytokine IL-2, suggesting that HCV inoculation profoundly alters the Treg ability to proliferate even in-vitro. Another consequence of this unexpected IL-2 secretion would be an increased survival of Tregs in-vivo after HCV inoculation. Indeed, Jeffery et al., in an effort to develop a therapeutic approach to strengthen Tregs to treat autoimmune liver diseases, described that a very low dose of IL-2 promotes the survival of Tregs [43]. Recently, Yang et al. conducted studies in IL-4 KO mice and suggested the essential role of IL-4 in supporting Treg-mediated immune suppression. Indeed, IL-4 was associated with increased cell survival and more interestingly to granzyme expression, one key molecule of Tregs’ suppressive activity [44].

In this study, we also described that HCV inoculation significantly increased the Treg suppressive activity. We inquired this inhibitory activity by evaluating the expression of several markers related to Treg function, and showed an increased *IL-2RA (CD25), IL35 (EBI3), GZMB* and *TGF-β1* expression after HCV inoculation. Tregs also up-regulated the secretion of the immunosuppressive cytokine TGF-β1, which is in accordance with previous reports in HCV-infected patient describing an increase frequency of Tregs positively correlated to a decrease of the host-immune response [8]. Another interesting observation is that Tregs displayed an increased expression of *IL-12, IL-17, IL-21* and *IL-23* pro-inflammatory cytokines after HCV inoculation. At 48 h p.i, Tregs secrete larger amounts of IL-17 cytokine and this result is consistent with findings showing that IL-17-secreting T cells are enriched within the liver of HCV infected patients [45]. Interestingly, other studies have described Th17-like Tregs [46] as abnormal Tregs characterized by IL-17 expression playing a proinflammatory role [47]. Moreover, some studies also indicate that IL-17-producing human Tregs retain efficient suppressive function [48]. Taken together, our data clearly suggest that Tregs worsen the suppressive microenvironment and favor liver damages in HCV-infected patients.

Some studies have mentioned that HCV induces the expression of chemokines that attract regulatory T cells to the site of infection, suggesting that HCV promotes an immunosuppressive environment within the liver, which impairs the activity of antiviral immune response [21,22]. Nevertheless, little is known about the mechanisms by which HCV induces Tregs recruitment. We show in our study that the supernatant of infected PHH induced migration of Tregs, which was substantially greater when Tregs were previously inoculated with HCV. These data suggest that human hepatocytes could favor the recruitment of Tregs in HCV-infected patient. In addition, our data confirmed an increased secretion of CCL20 by infected PHHs and the addition of neutralizing antibodies against CCL20, to the supernatant of infected PHH significantly inhibited the Treg recruitment, suggesting a prior role of the CCL20 pathway. In addition, we observed an early and sustained over expression of the CCR6 (the CCL20 receptor) gene into inoculated Treg. Moreover, while PHH overexpressed CCL2, Tregs overexpressed its receptor CCR2. It has been shown that CCR2 enhances the expression of CD25 on Tregs and influences their migration to the CCL2 secreting site as well as enhancing its epithelial to mesenchymal transition and invasion at the site [49,50]. Thus, the CCR2/CCL2 axis is clearly involved in the specific process of migration of Treg in the presence of HCV. Taken together, our observations were consistent with studies showing an increased recruitment of natural Tregs to the liver in HCV-infected patients [22,51]. We also addressed whether HCV might influence the chemotactic potential of intra-hepatic fibroblasts (IHF), which are other important liver cell components [16,17,52]. Our results revealed no significant differences in the Treg recruitment between supernatants of inoculated IHF compared to non-inoculated IHF supernatants. This result is interesting because the permissivity of IHF to HCV is highly debated and their role in the hepatic pathology may not be linked to the immune response but to the increase of fibrosis. Taken together, our results show that the intra-hepatic Treg recruitment is essentially mediated by human hepatocytes that are the principal target cells of HCV infection [24].

In conclusion, our pilot study provided strong evidences that the HCV infectious particles had the potential to shape escapes from the immune response by driving intra-hepatic Treg migration, inducing Treg expansion and finally increasing Treg regulatory phenotype and suppressive function.

## Figures and Tables

**Figure 1 cells-08-01296-f001:**
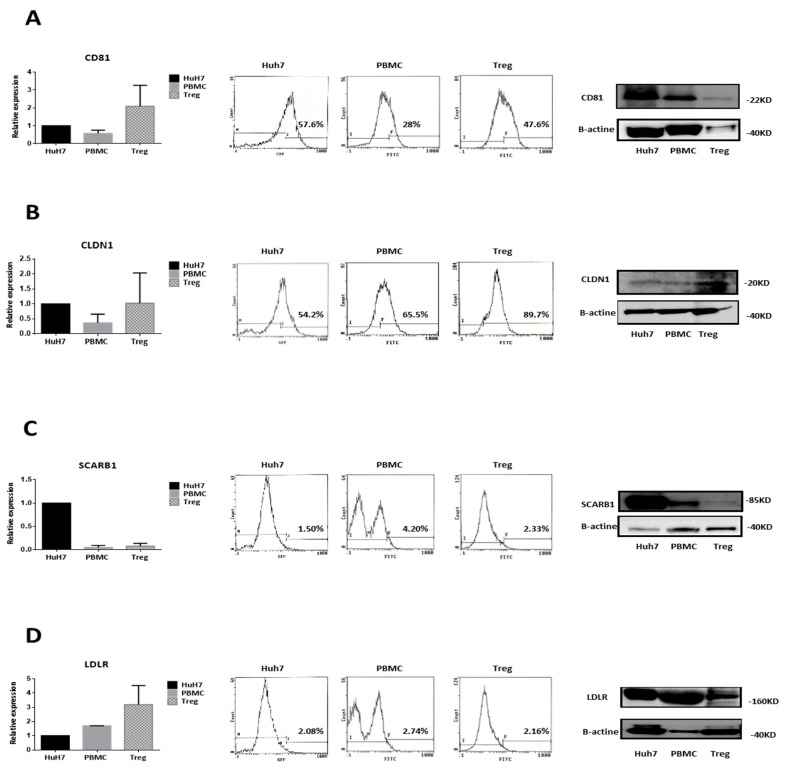
Tregs possess the classical hepatitis C virus (HCV) entry receptors: CD81, CLDN1 and LDLR. Gene expression analyses (qPCR) of receptors CD81 (**A**), CLDN1 (**B**), SCARB1 (**C**) and LDLR (**D**), associated with HCV entry in host cells (left panels). Results are expressed as relative expression and presented as means of four independent experiments ± standard error of the mean (SEM) bars. Protein expression Flow Cytometry (FACS) of receptors related to HCV entry (middle panels). Results are expressed in histograms displaying the percentage of cells positive for protein labeling compared to isotype control. Protein expression (Western Blot) of receptors linked to HCV entry (right panels). Analyses were performed on Huh7 (control), PBMC and Tregs. The images are representative of at least four independent experiments.

**Figure 2 cells-08-01296-f002:**
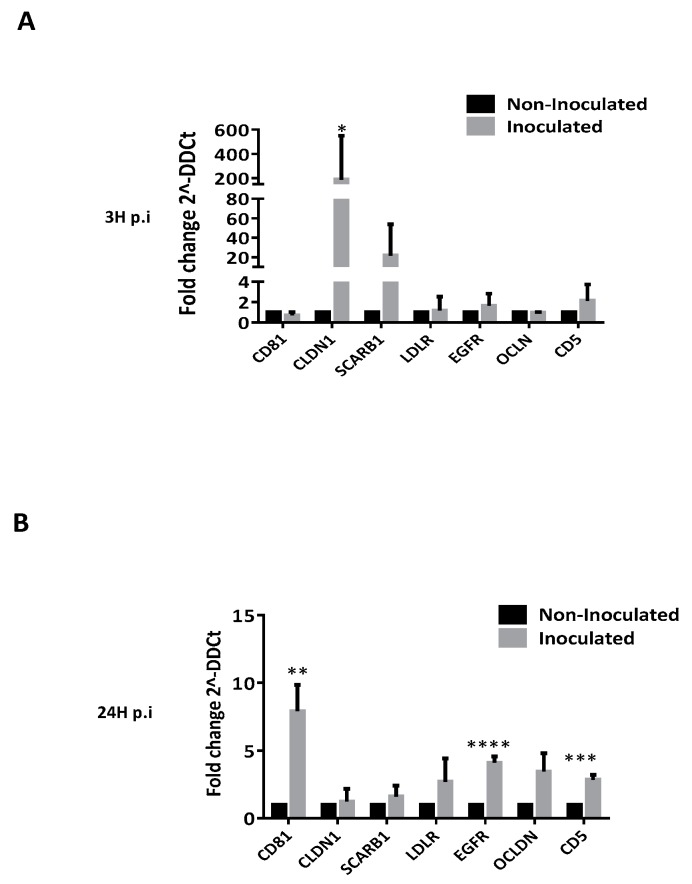
HCV inoculation increases the expression of its receptors on Tregs. HCV inoculation increases the mRNA expression of *CLDN1* and *SCARB1* at 3 h post-inoculation (3 h p.i) (**A**) and *CD81, EGFR* and *CD5* at 24 h p.i (**B**). Gene expression is normalized using GADPH, beta-actin, 18s and Hypoxanthine-guanine phosphoribosyltransferase (HRPT) mRNA as a housekeeping-gene before being reported to non-inoculated Tregs (black bars*).* Results represent means of five independent experiments and are presented as fold change (2^–ΔΔCt^) ± SEM bars. *p* ≤ *0.05* (*), *p* ≤ 0.001 (**), *p* ≤ 0.0001 (***) and *p* ≤ 0.00001 (****).

**Figure 3 cells-08-01296-f003:**
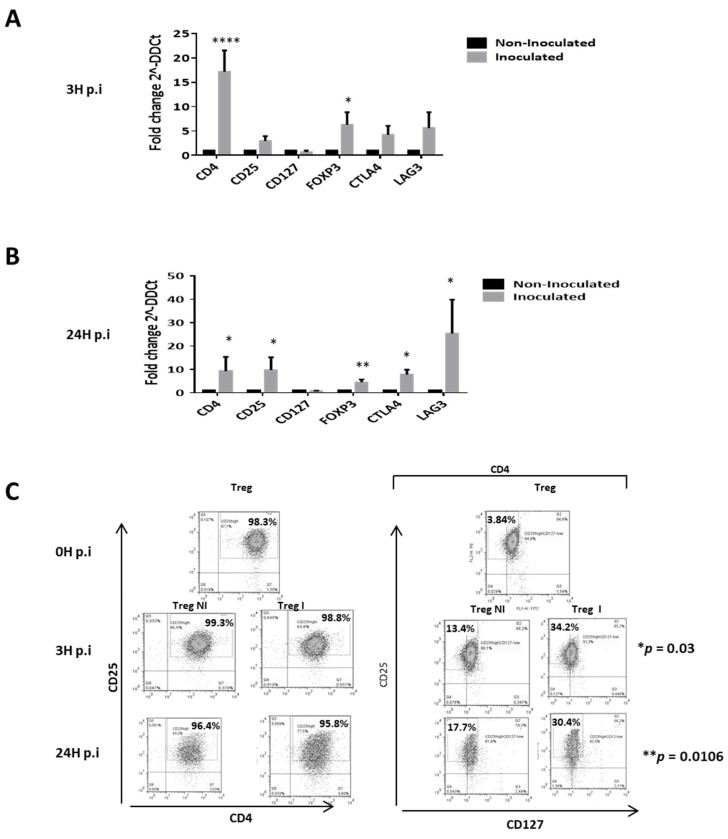
HCV inoculation increases the suppressive phenotype of Tregs. HCV inoculation affects CD4, CD25, CD127, FOXP3, CTLA4 and LAG3 expression in activated Tregs at 3 h post HCV inoculation (3 h p.i) (**A**) and 24 h p.i (**B**). Results are presented as means of four independent experiments of inoculated Tregs (light grey bars) versus non-inoculated Tregs (dark bars). Gene expressions are normalized by using GADPH, beta-actin, 18s and HRPT mRNA as housekeeping-genes and result are expressed in fold change (2^–ΔΔCt^) ± SEM bars. (**C**) Representative dot plot of double stained CD4+CD25^high^ and CD25^high^ CD127^-/low^ Tregs 0 h post HCV inoculation (0 h p.i), 3 h p.i and 24 h p.i.

**Figure 4 cells-08-01296-f004:**
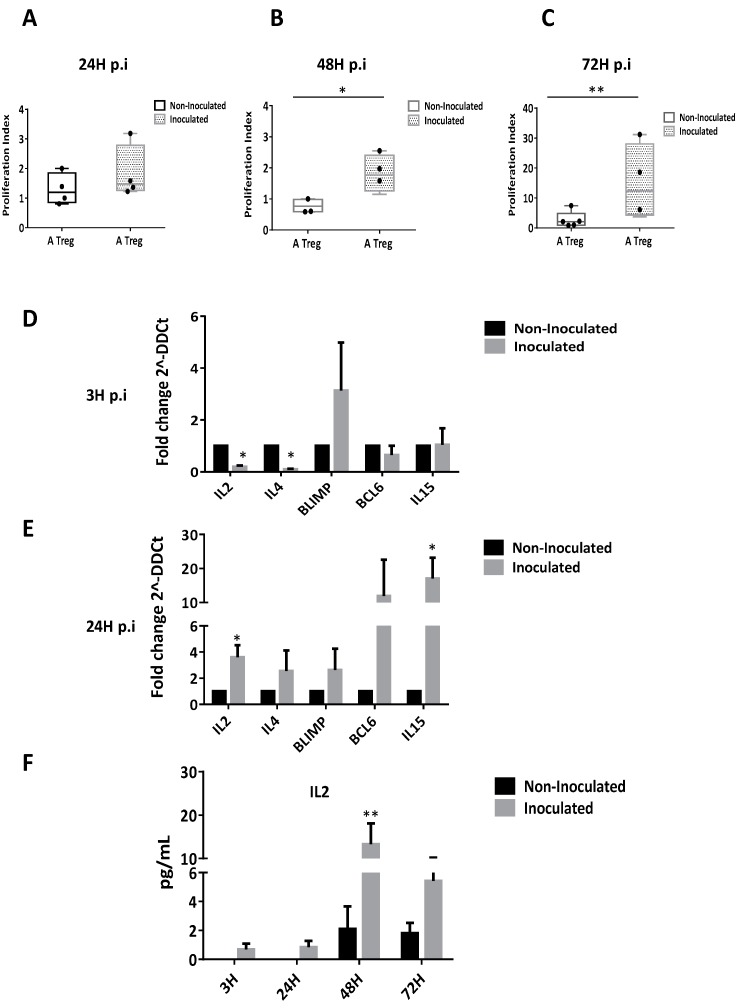
HCV inoculation increases the in vitro proliferative capacity of Tregs. Tregs proliferation was measured by using [^3^H]-thymidine incorporation assay and cell counting assays at 24 h (**A**), 48 h (**B**) and 72 h (**C**) after HCV inoculation. Results are expressed in index of proliferation of inoculated Tregs (hatched column) compared to non-inoculated (light column). Results are presented as mean values of triplicate ± SEM bars of four independent experiments. qPCR analyses showed that HCV inoculation increases *IL-2, IL-4, BLIMP1, BCL6* and *IL-15* gene expressions in Tregs at 3 h p.i, (**D**) and 24 h p.i (**E**). Results are presented as means of four independent experiments in inoculated Tregs (light grey bars) versus non-inoculated Tregs (dark bars). Gene expressions are normalized by using GADPH, beta-actin, 18s and HRPT mRNA as housekeeping-genes and the results are expressed in fold change (2^–ΔΔCt^) ± SEM bars. Secretion of the proliferative cytokine IL-2 by Tregs after HCV inoculation (**F**). Results are expressed as mean of three independent experiments and presented in pg/mL ± SEM bars comparing secretion by inoculated Tregs (light grey bars) versus non-inoculated Tregs (dark bars). *p* ≤ 0.05 (*), *p* ≤ 0.001 (**).

**Figure 5 cells-08-01296-f005:**
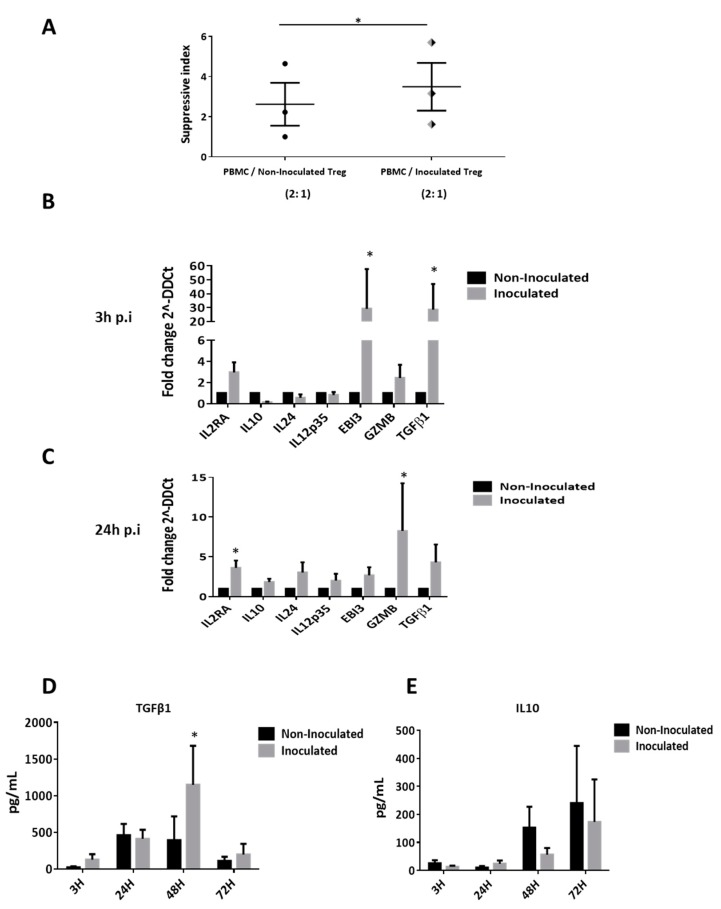
HCV inoculation increases the suppressive activity of Tregs. Tregs were pre-incubated for 24 h with HCV virions produced in cell culture (HCVcc) before co-cultured with autologous PBMC at 2:1 ratio in activated conditions (**A**). Results are expressed as mean values of triplicates of three independent experiments and presented in index of suppression ± SEM bars. Mechanistic analyses highlight an increase expression of IL-2RA, IL-10, IL-24, IL-12p35, EBI3, GZMB and TGF-β1 mRNA in inoculated Tregs at 3 h p.i (**B**) and 24 h p.i (**C**). Results are presented as means of four independent experiments in inoculated Tregs (light grey bars) versus non inoculated Tregs (dark bars). Gene expressions are normalized by using GADPH, beta-actin, 18s and HRPT mRNA as housekeeping-genes and the results are expressed in fold change (2^–ΔΔCt^) ± SEM bars. Secretion of immunosuppressive cytokine TGF-β1 (**D**) and IL-10 (**E**) was evaluated at different time point after HCV inoculation on three independent experiments. Results are expressed in pg/mL ± SEM bars comparing secretion by inoculated Tregs (light grey bars) versus non-inoculated Tregs (dark bars). *p* ≤ 0.05 (*).

**Figure 6 cells-08-01296-f006:**
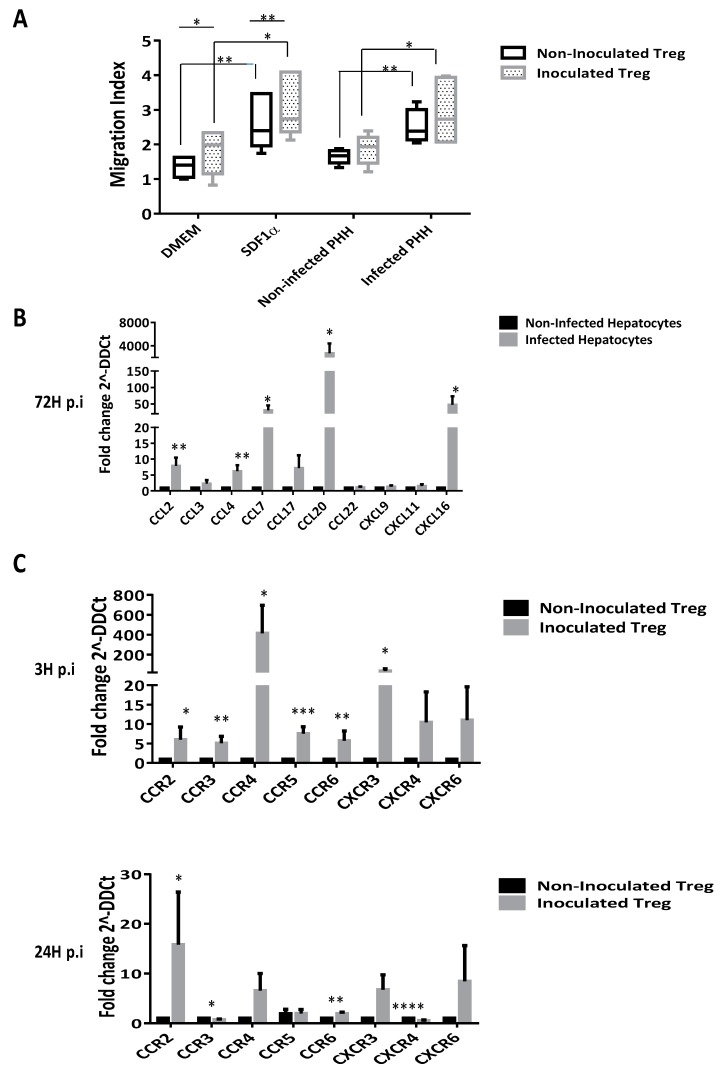
Tregs are recruited by infected primary human hepatocytes supernatants. Tregs recruitment by primary human hepatocytes (PHH) was evaluated by a Boyden chamber assay. In vitro infected PHHs recruit more natural Tregs than non-infected PHHs (**A**). Results are expressed as mean values of quintuplicate of six independent experiments and presented in index of migration of Tregs ± SEM bars. qPCR analyses show an increase of the expression of several chemokines associated with Treg recruitment such as CCL20, CCL17 or CXCL16 by PHH (**B**). Results are presented as means of five independent experiments on infected PHH (light grey bars) versus non-infected PHH (dark bars). HCV inoculation also increases the expression of corresponding chemokines receptors by isolated Tregs at 3 h p.i and 24 h p.i (**C**). All gene expressions are normalized by using GADPH, beta-actin, 18s and HRPT mRNA as housekeeping-genes and the results are expressed in fold change (2^–ΔΔCt^) ± SEM bars of five independent experiments. Secretion of three chemokines CCL20, CCL17 and CXCL16 has been examined by ELISA (**D**–**F**, left panels). Results are expressed as the mean of four independent experiments and presented in pg/mL ± SEM bars comparing secretion by infected PHH (light grey bars) versus non-infected PHH (dark bars). The chemotactic potential of PHHs on CD4+CD25^high^CD127^−/low^ Tregs was investigated in a Boyden chamber assay, using DMEM medium as a negative control; h-rec-CCL20, h-rec-CCL17 and h-rec-CXCL16 as positive controls and PHH supernatant with specific blocking anti-chemokines (**D**–**F**, right panels). Results are presented as means of quintuplicate of four independent experiments and presented in the index of migration related to DMEM medium ± SEM bars. When stated, statistical analysis were achieved by comparing the pointed condition to either infected PHH in the presence of non-inoculated (NI; $) or inoculated (I) Tregs (*), 50 ng of human recombinant (h) CLL20 with NI (+) or I (#) Tregs, h CCL17 in the presence of NI (•) or I (◦) Treg. *p* ≤ 0.05 (*), *p* ≤ 0.001 (**), *p* ≤ 0.0001 (***) and *p* ≤ 0.00001 (****).

**Figure 7 cells-08-01296-f007:**
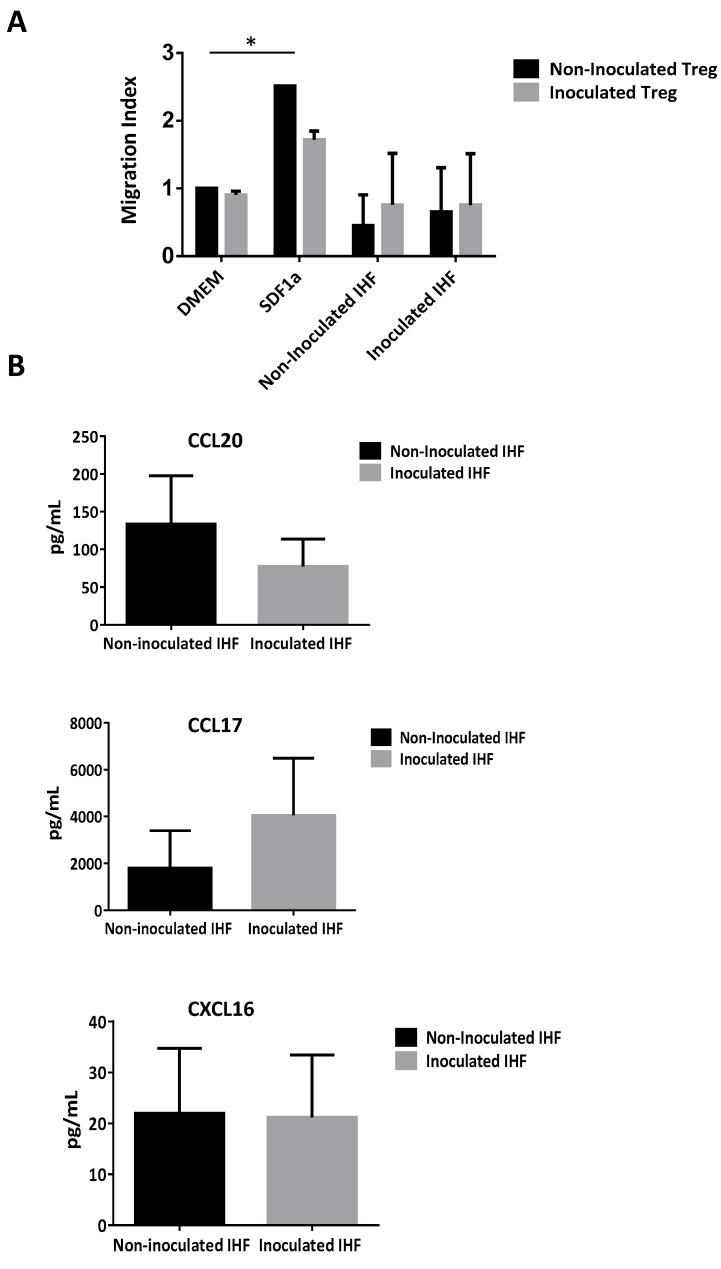
Tregs cannot be recruited by primary human intra-hepatic fibroblasts even upon HCV inoculation. Tregs recruitment by in vitro inoculated primary human intra-hepatic fibroblast (IHF) was evaluated by a Boyden chamber assay (**A**). Results are expressed as mean values of quintuplicate of two independent experiments and presented in an index of migration of Tregs ± SEM bars. Secretion of chemokines CCL20, CCL17 and CXCL16 has been examined by ELISA (**B**). Results are expressed as the mean of two independent experiments and presented in pg/mL ± SEM bars comparing secretion by inoculated IHF (light grey bars) versus non-inoculated IHF (dark bars). *p* ≤ 0.05 (*).

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
