# Peer review of "Hepatitis C Virus Improves Human Tregs Suppressive Function and Promotes Their Recruitment to the Liver"

_cells, 2019, doi:10.3390/cells8101296_

Round 1

Reviewer 1 Report

Review

Article by Ouaguia L et al presents interesting data on the effect of HCV cll culture produced virions (HCVcc) on the phenotype and functions of regulatory T cells, including potential infection of Tregs by HCVcc possible in principle due to finding respective receptors on Tregs.

Major

I see several major issues that authors need to address to make this article acceptable for publication.

Expression of potential rceptors for HCV on Tregs visualized by Western blotting (Fig 1 (left panels) needs to eb quantified for each experiment an presented also as results of this quantification. Same for FACS data – average values for % positive cells with STD have to be presented as supplement. In each experiment, mRNA levels, expression of receptors shown by FACS and protein expression levels by Western blotting need to be correlated. As for now, these three ways of measuring expression levels do not give concordant data. For example high CD81 expression on Tregs does not correlate to CD81 expression by Tregs when assessed by WB. Expression of LDLR by PBMC by mRNA does not correlate to WB data either, etc. All comparisons of the levels of expression have to be statistically verified. Authors state that coculturing Tregs with HCVcc increase expression of potential HCV receptors on these cells. This is not 100% true. Supplementation of cell cutlure medium from hepatocytes releasing HCVcc causes the effect. It can as well be due to complex factors, chemikones etc, released by infected hepatocytes in cell culture medium. To find what is the case, authors need to check the effect on Tregs of HCVcc as such. HCvcc can be purified by ultracentrifugation, and added to cell cutlure medium of Tregs in increasing concentration. Also, seasoned cell cutlure emdium from heaptocytes containing HCVcc can be filtered or ultracentrifuges to take away HCVcc and tested for effect on Tregs, also in dilutions with fresh medium. Authors state that treatment of Tregs with cell cutlure medium containing HCVcc increases expression of potential HCV receptors on Tregs – listing all. However, Fig 2 shows only few that are statistically significantly increased. SCARB1 at 3 is not – at least figure 2a does not shwo it. Text on page 9 decribing expression of cell surface molecules of Tregs characteristic to suppressive phenotype is very difficult to read, with all the references to cross-comparisons of different types of Tregs, untrated, treated with medium from uninfected hepatocytes, and infected hepatocytes. Basically, it repeats Table 2. This section has to be rewritten towards clarity. Importantly, it compared % stained cells – not clear, in one experiment, or as average. If it is avergae value, it has to be given with deviation, and also listed in Table 2, with statisitical comparison of average values. Otherwise, statements made of what is expressed more, or less, are not valid. Section 3.5 states that HCVcc effectively increases Treg proliferation (of suppressive subpopulation). At the same time, it states that transcriptomic analysis showed a decrease in expression of several molecules related to lymphocyte proloferation. This needs to be clarified. Figure 4 presents change of expression of proliferative markers and cytokines with time during HCVcc and Treg co-culture. It is unclear, why mRNA is measured only at 3 and 24 h, while cell proliferation and IL-2 levels - at 24, 48 and 72 h (IL2 also at 3h). High levels of proliferation are detected at 72 h. What is the reason for such different measurements, it should be motivated. Experimentally, measurement of T cell proliferation or suppression is not decribed at all. If increase in proliferation index, or loss of proliferation – suppression index, is less than 2, measurement can be considered as significant – two-fold difference is norallyu considered as a cut-off level. This relies to data in both figures 4 and 5. Number of measurements in the experiments (Fig 4ABC, and Fig 5A) is to few to make any conclusion, again, significance is not shown. Statements on lines 237 and 238 of legend for Fig 4 and 263-265 contain mistakes – do not reproduce what can be seen in the figures. Data in Fig 6 evokes questions. Less than 5-fold change in gene expression level cannot be considered as indicative of significant difference (relies to Fig 6C lower panel). Figure compares effects of HCVcc inoculation on migration of cells and relies it to expression of a panle of chemokines. For some reason, measurements of chemikine expression profiles for hepatocytes and Tregs are made at different time points, 72 h vs 3 and 24 h post inoculation, respectively. For hepatocytes, change in expression is recorder 72h post treatment, for Tergs – 3 and 24 h post treatment. These measurments have to be made at same time points, and additional data presented in supplement. Data in Fig 6 points at involvement of CCL20 in responsiveness of Tregs to HCVcc or HCVcc-treated hepatocytes. To prove this, authors treat Tregs with anti-CCl20 antibodies in different concentrations, and monitor the effect of treatment on cell migration. Antibodies to less involved or non-involved chemokines CCL17 and CXCL16 serve as a negative control. Positive control are infected PHH (assumed to secrete these chemokines), and chemokines added as proteins.

Interestingly, all added chemokines, also infected PHH (their cell cutlure medium),  increase migration of all Tregs, treated and not treated with HCVcc, indicating that Tregs are responsive to them. However, anti-CCL17 and anti-CXCL16 have no effect on migation. Increasing of concentration of anti-CCL20 antibody seem to decrease migation of Tregs, however the statistical significance of decrease is not shown. Altogether, these data question the specificity or the conditions of antibody blocking.

In my view, a control of the effect of cell culture medium from non-infected PHH on Tregs cells is needed, since the effects observed may be due to chemokines (cytokines, other factors) produced by PHH as such, not related to HCV (HCVcc) infection. Here, the only control for difference between uninfected and infected PHH as the amount of chemokines (CCl20, CCL17, CXCL16) they produce, and in NO case they show any statistical difference, indicating that chemokine production is not related to HCV (HCVcc) infection. Infected PHH appear not to produce more than unfected, and effect of their cell culture medium on Tregs cannot be HCVcc infection dependent.

Overall data lacks statistical evaluation.

Minor

What do authors mean when saying recruitment of intrahepatic Tregs – where to recruit, they are already in the liver (line 71), where the bulk of hepatocytes is HCV infected.

One cannot “increase.... cell phenotype” (line 74-75) – only expression of phenotype markers.

Data on Treg migration presented in panels A of Fig 6 and Fig 7 are presented in different ways – which makes hard to compared the effect on Treg recruitment of infected and noninfected (noninoculated) PHH and IFH. It is preferable to make plots that show how many measurements ingo into calculation of migration index (see actual values as dots).

Fig 3B, mistake in X axis – CD2127 – should be CD127

Fig 7 – IHF cells are called INOCULATED, and INFECTED – meaning theyr treatment with HCVcc containing cell cutlure medium from hepatocytes. One should adhere to one and the same term. Inoculation is correct one, since infection is not studied and proven.

MLR assays, Tr1 cells  line 67)– abbreviations not dechiphered.

Language

Article is very inaccurately written with lot of language mistakes, misspellings and statments unclear due to incoherent language, starting from the title and Abstract:

Promotes their arecruitment (line 3, title) In the progression hepatocellular carcinoma (line 21-22) Abilit of Tregs to hanf HCv on (line 23, slang) Expression of .... were monitored (lines 27-28)

Etc, needs extensive language revision.

Reviewer 2 Report

The authors presented new data to study the interaction between T regulatory cells and HCV. It was shown that Tregs possess the classical HCV entry receptors and HCV inoculation increases the expression of its receptors on Tregs; HCV inoculation increases the suppressive activity of Tregs; Tregs are recruited by infected primary human hepatocytes but not by intrahepatic fibroblasts.

The article can be published after correcting the section Results.

Key Notes

Introduction

Page 2, line 58: If there is a mention of induced Tregs, give their characteristic and references; line 67: briefly describe the function of TR1 - regulatory T cell type 1

Materials and Methods

Page 2, lines 92-94: Specify antibody for flow cytometry.

I recommend moving Table 1. Primer Sequences (page 4) to Supplementary Material and Methods.

Results

Page 9, lines 207-208: The sentence “…extensive analyses reveal that HCV inoculation increases  the expression of 22 markers related to Treg phenotype…” should be deleted, because results not shown.

Page 11, Table 1: How can you explain such a big difference between samples?

Page 12, Fig. 4 A, B, C - fix the labels on the x-axis.

Page 12, lines 237-238: the text does not match the data on the Fig. 4 D, E, taking into account the statistical significance. The same applies to Fig. 6 C and lines 285, 304; Fig.6 C and lines 287; Fig. 6 В-А and lines 289-290.

Page 13, lines 255-258: incorrect interpretation of data in Fig.S5: the concentration of IFN-γ does not change. HCV increases the expression / secretion of IL-17 and IL-23 only. Why do you associate pro-inflammatory cytokine IFN-γ with suppression?

It is necessary to edit the graphics in Fig. 6 D-F, right panel, because they are very difficult to understand. The effect of inhibition depends very little on the concentration of antibodies.

Reviewer 3 Report

Comments

1-    The manuscript is poorly written. English needs to be largely revised 

2-    The manuscript title : “Arecruitment” should be corrected

3-    Abstract, lines 27-28 : “Expression of Tregs phenotypic…and functional markers were monitored” should be “Expression of Tregs phenotypic…and functional.. markers was monitored”

4-    Abstract, line 29 :  “MLR” should be described clearly

5-    Introduction, line 44:  “Claudine1” should be Claudin-1”

6-    Introduction, line 44:   “LDLR” should be described clearly

7-    Introduction, line 47:  “Resolution and control of primary HCV infections is physiologically…” should be “Resolution and control of primary HCV infections are physiologically…”

8-    Introduction, lines 49-50:  “the development and persistence of a chronic infection is linked to…” should be “the development and persistence of a chronic infection are linked to…”

9-    Introduction, line 52:  “partly due…” should be “partly due to…”

10- Material and methods, line 125: “…as a specific controls” should be “…as specific controls”

11- Material and methods, line 131: “…different manufacturer” should be “…different manufacturers”

12- Description of β-actin should be corrected in the overall manuscript

13- Material and methods, line 181: “to makes…” should be “to make…”

14- Results, line 258:  “HCV not only improve Tregs phenotype but also increase….” Should be “HCV not only improves Tregs phenotype but also increases….”

15- Results, line 279: “the proportion and number CD4+CD25highCD127-Tregs was significantly…” should be “the proportion and number of CD4+CD25highCD127-Tregs were significantly…”

16- Discussion, line 328: “no study have addressed…” should be “no study has addressed…”

17- Discussion, line 336: “occludine” should be  “occludin”

18- Discussion, line 342: “These data correlates…” should be “These data correlate…”

19- Discussion, line 342: “HCV successfully bind…” should be “HCV successfully binds…”  

Round 2

Reviewer 1 Report

Manuscipt cells-559915 by L Ouaguia et al has been improved, but needs additional revision

as there are still many inaccuracies and unclarities.

Major comments

Critics from the 1st review #1 is met. Authors gave data for assessment of mRNA of HCV receptors on Tregs, and gave % of cell populations positive for these receptors by FACS (as Suppl to Fig 1).

In response to critical point #1, authors stated that their main point was not compare levels of expression of HCV receptors in different cell types, but just to prove that Tregs express these receptors. Honestly, the purpose of these experiments was mainly to show that the Tregs expressed the main receptors of HCV, without trying to show that the expression was more or less important in the Treg. I fully agree that this is important statement. However, as authors rightfully stated this in discussion, 70% of PBMCs are T cells. T cells are known to express all given receptors:

CD81 is a widely expressed protein, required for multiple normal physiological functions, expressed in many immune and non-immune cells, and required for cell fusion (nice review by S Levy 2014 https://www.ncbi.nlm.nih.gov/pubmed/24522698) . T cells do express CD81 (Levy S, 2014). It is actually illustrated also in this article – see Suppl Fig 3 panel A, - Tconv (CD25-) express CD81. Claudins (CLDN1 analysed in this study) are expressed by activated leukocytes, including APCs, B-cells and T-cells with increase in expression associated with immune activation. (https://www.ncbi.nlm.nih.gov/pmc/articles/PMC3822847/). Scavenger receptor BI (SR-BI or Scarb1) is a well-established HDL receptor, and was isolated in T and B cells from spleen and SR-BI expression regulates T and B cell proliferation (https://www.ncbi.nlm.nih.gov/pmc/articles/PMC3202973/ in mice). Human protein atlas shows expression in multiple cell types including lymphoid cells https://www.proteinatlas.org/ENSG00000073060-SCARB1/cell PBMCs express LDLR (https://www.ncbi.nlm.nih.gov/pmc/articles/PMC1905048/). LDLR is also overexpressed by stimulated human T cells  (see for example, https://www.ncbi.nlm.nih.gov/pmc/articles/PMC3196240/).

Tregs as subpopulation of T cells would then also express CD81, CLDN1, SRB1, LDLR  - just finding  these receptors on Tregs is not a novel. More data has to be presented – like quantitative analysis of the expression levels in the donors included in the study – like of the levels of receptors in donors PBMC to those in Tregs from the same donor, and/or comparison of the levels of expression of all four receptors in each donor.

Authors assessed Huh7 cells as positive control, PBMCs and Tregs. From these data one can see that all these cells express given receptors, i.e assessments do not have a negative control – non-expressing cells. Only on these background, one can state that Tregs have this unique expression quality – comparable to Huh7 (as stated) (while other cells do not have it – as in part is shown by authors for HepG2). It does not have to be primary donor cells, but could be cell lines. Plenty of non-expressing cells lines can be used as such based on the data on specific protein expression in different human cell lines available from Human Protein Atlas. Expression tests in Fig 1 require negative controls as confirmation of specificity of tests. Authors responded that expression levels by mRNA and other methods do not correlate. Anyway, as it is well described, mRNA levels are often not reflected in protein levels due to multiple layers of gene regulation like genomic variations, gene expression, protein translation and post‐translational modifications Dhirendra Kumar Gourja Bansal, Proteomics, 2016). This could explain some discrepencies as such observed for LDLR mRNA and protein expression. In addition, many studies reported discordant correlation between some subsets of protein and mRNA expression (Guoan Chen, Tarek G. Gharib et al, molecular and cellular proteomics,

2002 ; Vogel C, Marcotte EM. Nat Rev Genet. 2012 ; Lahtvee PJ1, Sánchez BJ, Cell Syst. 2017 ).

It was easier to run the correlation than to motivate why it may not work.  One can rank the levels of expression in Huh7, PBMC and Tregs based on four measurements presented in Suppl Tables 2 and 3 – 1 as the strongest, 3 as the lowest (see below). Ranked expression levels of all receptors correlate very well for PBMCs, and are less consistent for Huh7 or Tregs. Also, that they are consistent (independently of cell type) for most of the receptors, but not for SCARB1, which is good and proves the solidity of the data.

Cell type

CD81

WB 

FACS

CLDN

WB

FACS

SCARB1

WB

FACS

LDLR

WB

FACS

Huh7

2

1

3

3

1

3

2

3

PBMC

3

3

1

2

2

2

1

1

Tregs

1

2

2

1

3

1

3

2

It is very simplistic analysis, correlation analysis would have been better, and manuscript would win if authors perform it and conclude on inter-consistency of this block of data in results and in discussion.

More on Fig 1 and Supplementary tables 1-3. Difference in expression levels of receptors by either WB or FACS varies between 6 and 15%, which is OK, and allows to present levels of expression as average with STDV. However, each measurement includes a run of Huh7 cells as pos expression control. Difference in levels of receptor expression in these runs reaches 20%! How such discrepancy in behaviour of a cell line can be explained? It questions the accuracy of the measurements on the overall. Once more, it points at the necessity of negative control in each run. Difference in the levels of receptor expression in Huh7 cells in different runs has to be explained and variation of the levels addressed in the results. Main critical point that remains only partly met - Critical point #2 of the 1st review stated

“Authors state that coculturing Tregs with HCVcc increase expression of potential HCV receptors on these cells. This is not 100% true. Supplementation of cell cutlure medium from hepatocytes releasing HCVcc causes the effect. It can as well be due to complex factors, chemikones etc, released by infected hepatocytes in cell culture medium. To find what is the case, authors need to check the effect on Tregs of HCVcc as such. HCvcc can be purified by ultracentrifugation, and added to cell cutlure medium of Tregs in increasing concentration. Also, seasoned cell cutlure emdium from heaptocytes containing HCVcc can be filtered or ultracentrifuges to take away HCVcc and tested for effect on Tregs, also in dilutions with fresh medium”

Principally, it means that all the effects observed by the authors are due to treatment of Tregs with cell culture medium of HCVcc infected hepatocytes containing HCVcc and cytokines, chemokines and other factors released by infected hepatocytes. The effect was never ascribed to purely HCVcc. In response to review, authors state that due to leave of one of the authors of the paper, HCVcc cannot by any longer purified. OK, but authors can filter cell culture medium to deprive it of HCVcc while retaining all soluble factors and check the effect of these HCVcc-conditioned medium free of HCVcc. This was not done, and clearly would not be done even if reviewers insist.

With this NOT DONE, authors HAVE NO DIRECT PROOF that the effect they observe is due to HCV, or HCVcc as such, or HCV or HCVcc infection of Tregs. The effects they observe are due  to the addition of conditioned medium from HCVcc infected hepatocytes. This has to be corrected throughout the text, including abstract and discussion.

Critics from the 1st review #7 remains. Materials and methods are still incomplete. Article do not contain descriptions of: i) cell proliferation and cell suppression tests; ii) quantification of expression of cytokines such as IFNg, IL2, TGFb, etc on protein level (when given in results in pg/ml); iii) Western blotting for expression of any of the receptors (CD81 et al in Fig 1, or OCLN and EGFR in Suppl Fig 2) – how cell lysates were prepared, conditions for PAGE, blotting, and last but not least for antibody staining of membranes. Materials and methods have to be completed by the description of these methods or references to their description, and reagents used, specifically kits for cytokines, and antibodies used in WB. Critics from the 1st review # 10 is not met. It questioned the specificity or the conditions of antibody blocking. Instead of commenting on how the blocking was done, whether it was successful and specific, as compared to effect of control antibodies. Instead of answering the question and re-evaluating the data, authors wrote The specificity of the blocking antibody can be address by looking on the data sheets of the

antibodies used in these experiments: https://www.peprotech.com/fr/anti-human-mip-3-2 In the results part, authors have to comment on the success of the blocking procedures for CCL17, CCL20 and CXCL16 (Actually onlu CCL20 was a success on qualitative level (as authors mentioned in the revised text).

It is clear that the study was done on very few patients. All supplements show at most 4 donors, most probably, the same in all tests. Number of donors included in the study has to be given in Materials and methods. Results, Discussion and Abstract have to indicate that this is a pilot study done on a limited number of samples.

Minor

Supplementary Table 1 and Table 2 are the same tables. Huh7 measurements are used as control, and cannot be placed in the table in the Donor NN graph. Supplementary tables have to be reformatted.

Run 1

Run 2

Run 3

Run 4

Huh7

Huh7

Huh7

Huh7

Donor 1 PBMC

Donor 2 PBMC

Donor 3 PBMC

Donor 4 PBMC

Donor 1 Tregs

Donor 2 Tregs

Donor 3 Tregs

Donor 4 Tregs

By data in Suppl Fig 3, T regs do not express LDLR – same as HepG2 – their staining with specific antibody does not differ from staining with control IgG. Suppl Fig3 data differs from data in Fig. 1D (donor 1) Symbols used over bars in Fig 6 panels D E, F to demonstrate statistical significance – stars, dots, etc – are very confusing and difficult to understand for a reader, of what is compared to what. Normally one draws lines pointing what data seta are compared with p value over the line. Suppl Fig 2 A – Left panels show Huh7, PBMC and Tregs. Right bars show expression by WB in Huh7, Tconv and Tregs. Which cells were actually used – PBMC, or T cells (Tconv)? Suppl Fig B, left panel – Huh7, PBMC, Tregs. Right panel - WB assessment is done for HeLa, Treg A and TregNA – which cells are the last two? Why PBMC are not used? Huh7 should also be shopwn on WB as negative control. Suppl Fig 2 – figure legend. Not images were assessed by WB, but expression was assessed by WB and presented as images. Suppl Fig 2, please, provide actin staining for right panel of WB which currently shows only OCLN and EGFR. Supplementary need to include quantification of OCLN and EGFR expression by WB related to actin expression in each pf three samples, to be presented as additional panel. Suppl Fig 3 B – correct name of cells – HepG2 (not Hepg2) Suppl Fig 5 – does not show impact of Tregs on the development of liver inflammation after HCV inoculation. This is too much! It is an in vitro experiment. Authors do not have HCV, only HCVcc. Only effect of conditioned medium containing HCVcc is tested. Lastly, figures shows expression by “conditioned” Tregs of the inflammatory factors, just this.

Additional suggestions/comments

When discussing expression of IL-17 by Tregs (Suppl 5) one can refer to studies of the abnormal proinflammatory Tregs charaterized by IL-17 expression (Bansal SS et al, 2018), and to the fact that IL-17 expression is characteristic to Th17-like Tregs (Dahen T et al, BLOOD, 2012). This can specify Treg population involved in suggested Treg enhancement of HCV infection.

To add to the discussion of the data presented in Fig 5:

IL-2 promotes Treg survival (Jeffery HC et a, Clin Exp Immunol, 2017); IL-4 supports Treg mediated immune suppression and associates with granzyme expression (Yang WC, Frontiers Immunol 14, Nov 2017).

Language revision

Manuscript has to undergo professional language revision left after the revision, and even appearing after the revision. Just few examples:

Lines 53-54 - vigorous HCV-specific CD4+ and CD8+ T cells expansion [6][6], which are able to clear the virus by non-cytolytic and cytolytic effector functions – cells are able to clear the virus, but this is attribuited to the word “expansion” Line 65 – one of their main mechanism of action is due by – one of several should be “one of mechanisms”, and mechanism is due to, not “by” Lines 80-81 - the mechanisms by which HCV might induce the 80 recruitment and proliferation of Tregs are less known – Less then what? Could have said ”not known”, ”remain unknown” etc Lines 258-259 – has to be corrected, “In contrast, at 24H p.i, there was an increase of all the proliferative factors especiallybut only significantly for IL-2 and IL-15 by inoculated Tregs (Figure 4E)” Currently it is meaningless. Lines 290-291 – “These data clearly show that HCV not only improves Tregs phenotype but also increase its suppressive and harmful activity” One cannot improve T cell phenotype! Needs editing. Lines 388-389 - These data correlate with studies showing that HCV successfully binds PBMCs into which the vast majority of cells are T lymphocytes – should be “of which, the majority are T cells”.
